# Tracking district-level performance in the context of achieving zero indigenous case status by 2027

Chander Prakash Yadav[1,2,3☯], Syed Shah Areeb Hussain [1,2☯], Sanjeev Gupta[1], Praveen K. Bharti[1,2], Manju Rahi[1,2,4], Amit Sharma [1,2,5]*

1 ICMR-National Institute of Malaria Research, New Delhi, India, 2 Academy of Scientific and Innovative Research (AcSIR), Ghaziabad, UP, India, 3 ICMR-National Institute of Cancer Prevention and Research, Noida, UP, India, 4 Indian Council of Medical Research, New Delhi, India, 5 Molecular Medicine Group, International Centre for Genetic Engineering and Biotechnology, New Delhi, India

☯ These authors contributed equally to this work.
* amit.icgeb@gmail.com

## Abstract

India has committed to zero indigenous malaria cases by 2027 and elimination by 2030. Of 28 states and 8 union territories of India, eleven states were targeted to reach the elimination phase by 2020. However, state-level epidemiology indicates that several states of India may not be on the optimum track, and few goals set in National Framework for Malaria Elimination (NFME) for 2020 remain to be addressed. Therefore, tracking the current progress of malaria elimination in India at the district level, and identifying districts that are off track is important in understanding possible shortfalls to malaria elimination. Annual malaria case data from 2017–20 of 686 districts of India were obtained from the National Center for Vector-Borne Diseases Control (NCVBDC) and analysed to evaluate the performance of districts to achieve zero case status by 2027. A district's performance was evaluated by calculating the annual percentage change in the total number of malaria cases for the years 2018, 2019 and 2020 considering the previous year as a base year. The mean, median and maximum of these annual changes were then used to project the number of malaria cases in 2027. Based on these, districts were classified into four groups: 1) districts that are expected to reach zero case status by 2027, 2) districts that would achieve zero case status between 2028 and 2030, 3) districts that would arrive at zero case status after 2030, and 4) districts where malaria cases are on the rise. Analysis suggest, a cohort of fifteen districts require urgent modification or improvement in their malaria control strategies by identifying foci of infection and customizing interventions. They may also require new interventional tools that are being developed recently so that malaria case reduction over the years may be increased.

**Data Availability Statement:** Malaria data for year 2017 and 2018 are in public domain that can be accessed from the link given below: District wise malaria data for 2017 (https://nvbdcp.gov.in/Doc/

Annual-Report-2017.pdf) and District wise malaria data for 2018 (https://nvbdcp.gov.in/Doc/Annual-Report-2018.pdf). Malaria data for 2019 and 2020 are not available online at present, but can be requested from National Center for Vector Borne Diseases Control (NCVBDC): National Center for Vector Borne Diseases Control (NCVBDC) 22, Sham Nath Marg, Delhi - 110054 Website: www.nvbdcp.gov.in/ E-MAIL: nvbdcp-mohfw@nic.in.

**Funding:** The authors received no specific funding for this work.

**Competing interests:** The authors have declared that no competing interests exists.

## Introduction

Malaria still causes considerable morbidity and mortality in many parts of the world. As per the World Malaria Report 2021, there were an estimated 241 million cases and an expected 627,000 deaths from malaria in 2020, with African countries estimated for 95% of all cases. The World Health Organization's (WHO) South-East Asia Region (SEAR) accounted for ~2% of all malaria cases globally. Malaria incidence in SEAR has fallen dramatically during the last two decades, from an estimated 23 million cases in 2000 to 5 million cases in 2020. Although India had the greatest absolute decline in the SEAR, from ~20 million estimated cases in 2000 to ~ 4.1 million estimated in 2020, it still accounted for an estimated 83% of malaria cases and 82% of malaria fatalities in the region [1]. Between 2001 and 2015, a significant expansion of malaria interventions comprising vector control, chemoprevention, diagnostic testing, and treatment resulted in a 30% reduction in malaria incidence worldwide and a 47% decrease in malaria fatality rate thus avoiding an estimated 4.3 million deaths [2, 3]. The World Health Organization encouraged this remarkable accomplishment and adopted the Global Technical Strategy for malaria 2016–2030 (GTS) in 2015 to provide technical support to countries in scaling up malaria responses and working towards elimination by lowering the global malaria burden by 90%, and to interrupt malaria transmission in at least 35 countries by 2030 [2].

In 2020, the WHO noted that global malaria morbidity and mortality were off-track from the 2020 targets for a reduction in malaria cases and deaths by 37% and 22% respectively [4]. The estimated malaria case incidence in 2020 was 56 cases per 1000 population, which was significantly higher than the targeted 35 cases per 1000 in GTS 2015, and only 3 cases per 1000 population lower than 2015 estimates [4]. The WHO predicted that if current trends continue, the 2030 targets for global malaria morbidity would be missed by 87% of malaria case incidence per 1000 population. Of the 92 malaria-endemic countries of 2015, only 31 were on track to achieve the GTS 2020 milestone for malaria morbidity (>40% reduction in malaria morbidity), 21 countries achieved reductions below the GTS targets, 9 countries did not achieve any significant change and 31 countries witnessed an increase in case incidence [4].

Having achieved an 84% reduction in reported malaria cases [5] and a 65% reduction in estimated case incidence between 2015 and 2020 [1], India was the only High Burden High Impact (HBHI) country to register a drop in malaria case incidence in 2020. However, service disruptions during the COVID-19 pandemic may have hampered India's progress toward malaria elimination, and the rate of reduction has decreased in the last couple of years [1]. To plan and coordinate the elimination strategy, the National Center for Vector Borne Diseases Control (NCVBDC) in India developed the national framework for malaria elimination 2016–2030 (NFME) and stratified states and union territories into Category 1 –Prevention of re-establishment phase (Zero malaria API), Category 2 –Elimination phase (State malaria API<1, and all districts having API<1), Category 2 –Pre-elimination phase (State malaria API<1, but some districts having API>1) and Category 3 –Intensified control phase (State malaria API > = 1) [6]. The NFME resolved to eliminate malaria from 15 Category 1 states by 2020, 11 Category 2 states by 2022, and the entire country, including Category 3 states by 2027 while maintaining zero transmission until 2030 [6]. All the nine states i.e., Goa, Haryana, Himachal Pradesh, Kerala, Manipur, Punjab, Rajasthan, Sikkim, and Uttarakhand, and six union territories i.e., Chandigarh, Daman Diu, Delhi, Jammu and Kashmir, Lakshadweep, and Puducherry in Category 1 fell short of their goal of having no indigenous malaria cases by 2020. A total of 13 of these states/union territories reported a significant reduction in cases (24,134 in 2014 to 2,146 in 2020), but Delhi and Lakshadweep witnessed an increase in malaria cases. Of the 11 states in category 2, i.e., Andhra Pradesh, Assam, Bihar, Gujarat, Karnataka, Maharashtra, Nagaland, Tamil Nadu, Telangana, Uttar Pradesh, and West Bengal, only eight states achieved

the NFME target of reducing malaria API below 1 case per 1000 population in all districts by 2020. Based on the NFME and with support from the WHO GTS 2016–2030, India formulated the National Strategic Plan (NSP) for Malaria Elimination 2017–2022 in which malaria was identified as a local-focal problem, and districts were selected as the operational units for planning and implementation of intervention strategies. Henceforth, a more granular stratification was adopted and districts were classified as Category 0 –Prevention of re-establishment (without local transmission and reporting no case for the last 3 years), Category 1 –Elimination (API < 1), Category 2 –Pre-elimination ($1 \leq$ API $\leq 2$), and Category 3 –Intensified control phase (API > 2) [7].

Despite the significant progress made in recent years, district-level malaria epidemiology indicates that it is likely that some of the goals set in the NSP may remain challenging. Therefore, the primary objective of the study was to project a year of zero case status for each of the selected districts by studying changes in recent (i.e. 2017–20) caseload reduction. This study has identified districts that are likely to fall short of zero case status by 2027. To set goals for elimination, the required rate of reduction in malaria cases to achieve zero indigenous cases by 2027 in these districts was also estimated. The results of this study can guide India to plan and scale up intervention efforts in vulnerable districts that are likely to fall short of the malaria elimination goals.

## Methodology

### Data analysis

Annual malaria cases data from 2017 to 2020 of 686 districts of India were obtained from the National Center for Vector-Borne Diseases Control (NCVBDC) and analysed to assess the performance of all districts towards achieving zero indigenous case status by 2027. The year 2017 was selected as the starting year of the analysis because India had significantly scaled up intervention efforts after the rollout of NSP (2016) activities shifting the programme efforts from control to elimination, which may not be accurately represented if malaria incidence data before 2017 would be used. Districts that already have attained zero case status before 2020 and the districts where average cases in the last four years (2017–20) are below 50 were excluded from the analysis as they are in the control phase and require specialized modelling which will require district-specific intervention data that are not available presently. In this study, our focus was on high malaria burden districts as they are more crucial in the present scenario for achieving malaria elimination by 2027.

### Districts performance in terms of malaria caseload reduction

The performance of a district was evaluated by calculating the Annual Rate of Change (AROC) in the total number of malaria cases for the years 2018, 2019, and 2020 considering the previous year as a base year by using the following formula:

$$\text{Annual Rate of Change (AROC) in malaria cases for } 2018 = \frac{\textit{Malaria cases in } 2018 - \textit{Malaria cases in } 2017}{\textit{Malaria cases in } 2017} * 100$$

$$\text{Annual Rate of Change (AROC) in malaria cases for } 2019 = \frac{\textit{Malaria cases in } 2019 - \textit{Malaria cases in } 2018}{\textit{Malaria cases in } 2018} * 100$$

$$\text{Annual Rate of Change (AROC) in malaria cases for } 2020 = \frac{\textit{Malaria cases in } 2020 - \textit{Malaria cases in } 2019}{\textit{Malaria cases in } 2019} * 100$$

Once the AROC was calculated, the Mean Annual Rate of Change (Mean AROC), Median Annual Rate of Change (Median AROC), and Maximum Annual Rate of Change (Max. AROC) were calculated on these values as given below using Microsoft Excel.

Mean Annual Rate of Change (Mean AROC) = Average (AROC in 2018, AROC in 2019, AROC in 2020)

Median Annual Rate of Change (Median AROC) = Median (AROC in 2018, AROC in 2019, AROC in 2020)

Maximum Annual Rate of Change (Mean AROC) = Maximum (AROC in 2018, AROC in 2019, AROC in 2020)

Mean AROC signifies the average performance of a district over the last three years. Since there were many districts with a significant variation in annual percentages across three years, Median AROC was also calculated to deal with higher variation. Maximum AROC represents the best performance of a district in three years. Based on different AROC (mean, median, and maximum), all analysed districts of India were classified into four categories: 1) districts that would achieve zero case status by 2027, 2) districts that are expected to achieve zero cases status between 2028 and 2030), 3) districts that would achieve zero case status after 2030, and 4) districts where cases are on the increase in the last three years. Further, assuming the best-case scenario, a list of districts has been identified which may miss the malaria elimination target if necessary actions/interventions are not deployed in a timely and efficient manner. The projection of malaria cases in the year 2027 for the district was done using the below-given approach:

Suppose we have a district named 'A' where the annual rate of change (AROC) in 2018, 2019, and 2020 were -23.56%, -31.51%, and -25.40% (it can be calculated as discussed above) respectively, and total cases in 2020 were reported as 6734. The mean AROC, median AROC, and Maximum AROC can be calculated as using MS Excel

Mean AROC = Average (23.66, 31.51, 25.40) = 26.83

Median AROC = Median (23.66, 31.51, 25.40) = 25.40

Maximum AROC = Maximum (23.66, 31.51, 25.40) = 31.51

If we consider the best-case scenario, 31.51% will be the Annual Rate of Change (AROC). Based on this rate of change, the projected cases in 2027 can be calculated as

| Cases in 2020 | : | 6734 |
|---|---|---|
| Cases in 2021 | : | $6734 - 6734 * \frac{31.51}{100}$ = 4612 (i.e. 31.51% reduction) |
| Cases in 2022 | : | $4612 - 4612 * \frac{31.51}{100}$ = 3159 |
| Cases in 2023 | : | $3159 - 3159 * \frac{31.51}{100}$ = 2163 |
| Cases in 2024 | : | $2163 - 2163 * \frac{31.51}{100}$ = 1482 |
| Cases in 2025 | : | $1482 - 1482 \frac{31.51}{100}$ = 1015 |
| Cases in 2026 | : | $1015 - 1015 * \frac{31.51}{100}$ = 695 |
| Cases in 2027 | : | $695 - 695 * \frac{31.51}{100}$ = 476 |

The above calculation can be rewritten in a simplified form using the compound interest formula given below:

$$\textit{Projected malaria cases in } 2027 = \textit{Malaria cases in } 2020 \times (1 + r)^7$$

Where '$r$' is the AROC (based on mean, median, or maximum) and 7 is the time component (from 2020 to 2027). All values were rounded off to 0 decimal places to denote whole cases. For example, projected Malaria cases in 2027 in district A will be $6734*(1-0.3151)^7 = 476.06174{\sim} = 476$

Similarly, the total number of years required to achieve zero case status for each of the districts can be estimated using a modified form of the compound interest equation (specified below), which was added to the base year (2020) to determine the projected year of zero case status as.

$$\textit{Projected year of zero case status} = \mathbf{2020} + \frac{\mathbf{\ln(1/C)}}{\mathbf{\ln(1+r)}}$$

Where '$C$' is the total number of cases in the base year (2020 in our case), and '$r$' is the AROC based on the mean, median or maximum value as defined above. All values were rounded off to 0 decimal places to obtain the final year of elimination. Finally, the required rate of change in the districts that are projected to not achieve zero case status by 2027 was calculated using the formula given below

$$\textit{Required annual rate of reduction} = \sqrt[^{(2027-2020)}]{\mathbf{1/(\textit{Total no. of cases in base year } (2020)}}$$

All analyses were carried out in Microsoft Excel and Stata 15.0. Geographical mapping was done using ArcGIS software.

## Limitations

The above analysis has certain limitations which are inherent to such work. In this study, the year of achieving zero status was not predicted by developing a statistical model using advanced statistical methods (such as Generalised Liner Model (GLM), time series regression, smoothing techniques or machine learning method), as they require larger volumes of data on multiple variables (such as environmental parameters: rainfall, humidity, temperature, vegetation, differential malaria control interventions, climate zones, etc). These were not available at a granular level for all districts of India. Since no formal statistical model has been formed to predict a year of zero case status, so no validation and sensitivity analysis has been performed. Rather we have projected the number of years to reach zero case status by studying the recent progress in malaria caseload reduction via extrapolation. The data used for this study is from the public sector only and misses private sector malaria cases. The malaria surveillance in the country uses the network of health care services from primary to tertiary level and the data are collated from grassroots (village) to tertiary (district) level. However, the surveillance of the Indian malaria control programme captures ~8% of the total caseload as per World Malaria Report 2017. A substantial case burden is in the private sector (both formal and informal) which does not get reflected in the national figures [8, 9]. A large unknown caseload in the private sector could influence our findings and impact the accuracy of our analysis.

## Results

Of the 686 districts in India that were analysed, 117 districts had already achieved zero case status by 2020 or before and were therefore excluded from further analysis (Table 1 and Fig 1). From the remaining 569 districts, 205 districts reported, on an average, less than 50 cases of malaria between 2017 and 2020 and were also excluded from further analysis, assuming that these districts are likely to be on the last leg of malaria elimination. The majority of districts that have zero malaria cases or less than 50 cases on average between 2017 and 2020 belong to

the northern states/UT of Jammu and Kashmir, Himachal Pradesh and Uttarakhand, north-eastern states of Sikkim, Arunachal Pradesh, Manipur and Nagaland and the southern state of Tamil Nadu. Along with these, many districts from Bihar and Uttar Pradesh reported very few malaria cases, however, whether the surveillance in them was robust remains to be assessed [10].

The projected malaria cases in 2027 were estimated for the remaining 364 districts using the last three years' annual rate of change data. Firstly, the year of zero cases status for each of the remaining 364 districts was projected using mean AROC, representing the average case-load reduction performance. Based on this approach, 202 districts are projected to achieve zero case status by 2027, and an additional 42 districts may achieve zero case status between 2028 and 2030 (prevention of re-establishment phase). However, 89 districts are projected to be unable to achieve zero case status by 2030, whereas in an additional 31 districts, the mean AROC between 2017–2020 is positive, i.e., on an average, they have witnessed a year-on-year increase in malaria incidence. Therefore, the year of achieving zero cases status is hard to predict unless these districts can resume reduction in malaria incidence (Fig 1 and Table 1). There are many districts where there was considerable variation in the annual reduction rate over the three years (2018–20). To overcome this, the median AROC was also calculated to project the zero case status of each district. Median AROC again represents average performance in terms of caseload reduction adjusted for extreme value. The median AROC suggested that 216 districts are on the right track and would be able to attain zero case status by 2027, and 46 districts are projected to achieve zero case status during the prevention of re-establishment phase (between 2028 and 2030). Of the remaining 102 districts predicted to miss achieving zero malaria cases by 2027, 19 districts have a positive AROC, indicating that malaria in these districts is on the rise though caseloads are low. As discussed above, it is difficult to project the year when these districts will achieve zero case status until they resume reduction (Fig 1 and Table 1). The majority of the districts that are predicted to miss the target using mean AROC and median AROC are located in the states of Odisha, Chhattisgarh, Uttar Pradesh, Jharkhand, West Bengal, Gujarat and Rajasthan. Moreover, most of the districts in Tripura and Mizoram and some in Maharashtra, Odisha, Uttar Pradesh, and Punjab have witnessed an increasing trend in malaria (Fig 2).

Finally, the year of zero case status was also calculated using maximum AROC, representing the best-case scenario as it works on the maximum reduction attained by a district in the last three years. Under the best-case scenario, 335 districts were projected to achieve zero-case status by 2027. Twenty-nine districts may achieve zero malaria case status post-2027. Amongst 29 districts, 14 may achieve zero case status between 2028 to 2030 (prevention of re-establishment phase), and the remaining 15 are very crucial because they need to go beyond 2030 to achieve zero malaria case status if necessary interventions are not done on time (Table 1, Figs 1 and 2). Districts projected to fall short of the zero case status target by 2030, even under the best-case

**Table 1. Summary of districts projected to achieve/not achieve zero case status using different methods.**

|  | Mean AROC | Median AROC | Maximum AROC |
|---|---|---|---|
| No. of districts that have already achieved zero case status | 117 | 117 | 117 |
| No. of districts with residual cases of malaria (1–50) | 205 | 205 | 205 |
| No. of districts projected to achieve zero cases status by 2027 | 202 | 216 | 335 |
| No. of districts projected to achieve zero cases status by 2028–2030 | 42 | 46 | 14 |
| No. of districts projected to achieve zero cases status after 2030 | 89 | 83 | 15 |
| No. of districts with increasing incidence of malaria | 31 | 19 | 0 |

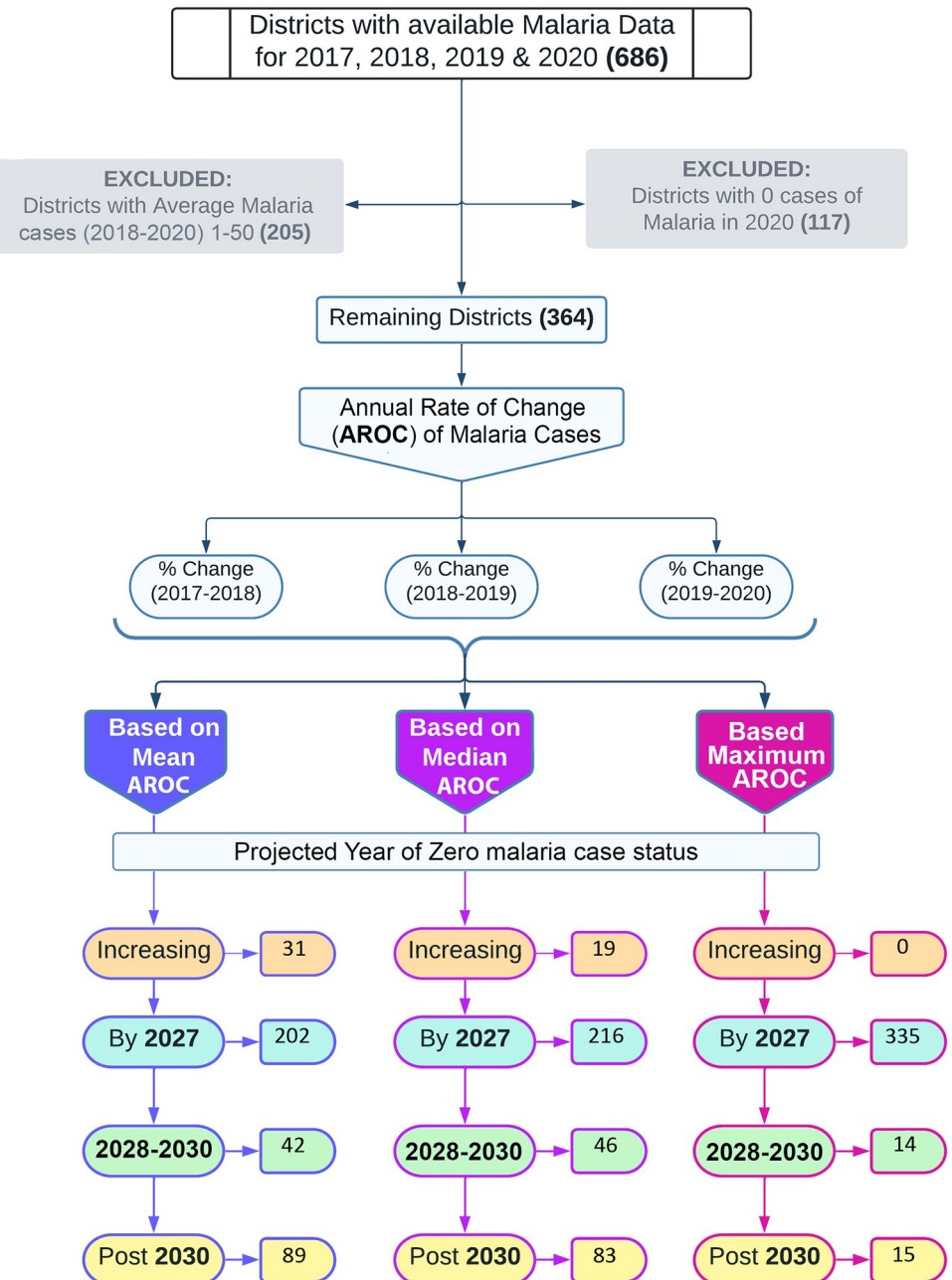

**Fig 1. Flow chart representing the classification of districts based the projected year of achieving zero case status of malaria using on three methods: Mean AROC, median AROC and maximum AROC.**

scenario, are the most vulnerable and need intense focus so that India can eliminate malaria by 2030. The fifteen districts that fall under this category are named as Dantewada, Bijapur, Narayanpur & Bastar in Chhattisgarh; West Singhbhumi in Jharkhand; Shahdol in Madhya Pradesh; Greater Mumbai and Gadhchiroli in Maharashtra; Lawangtlai & Lunglei in Mizoram; Junagadh in Gujarat; Badradri in Telangana; West Tripura in Tripura; Badaun in Uttar Pradesh and Kolkata in West Bengal (Fig 2). Amongst these districts, Greater Mumbai has the lowest AROC and the highest number of projected cases in 2027 under the best-case scenario

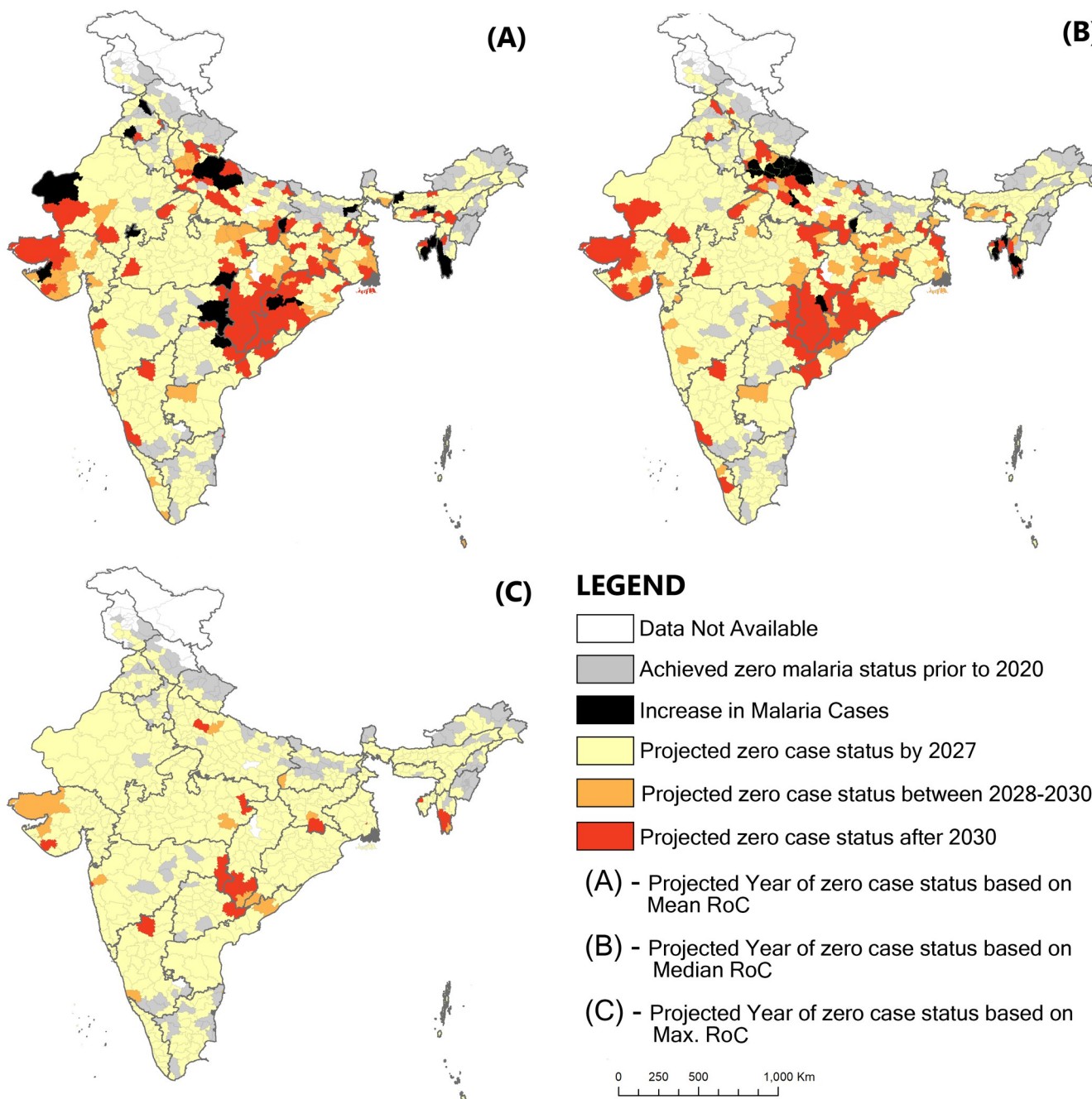

**Fig 2.** Classification of Indian districts as per projected year of achieving zero case status using three approaches: (A) mean AROC, (B) median AROC and (C) maximum AROC.

(Fig 3). At the present pace, it is projected to report as many as 1469 malaria cases in 2027 (Fig 3), and an increment of more than 59% in AROC is needed to be able to achieve zero case status by 2027 (Fig 4). A high increment in the AROC is also needed in the districts of Lawangtlai (48.8%), Lunglei (47.8%), Dantewada (45.5%), Junagadh (39.6%), Kolkata (38.4%), Badaun (38%), Narayanpur (33.6%), Bijapur (34.7%), West Tripura (32%), Bastar (31.7%) and West Singhbhum (31%) (Fig 4). Other than these, Badaun (Uttar Pradesh) and Kolkata (West

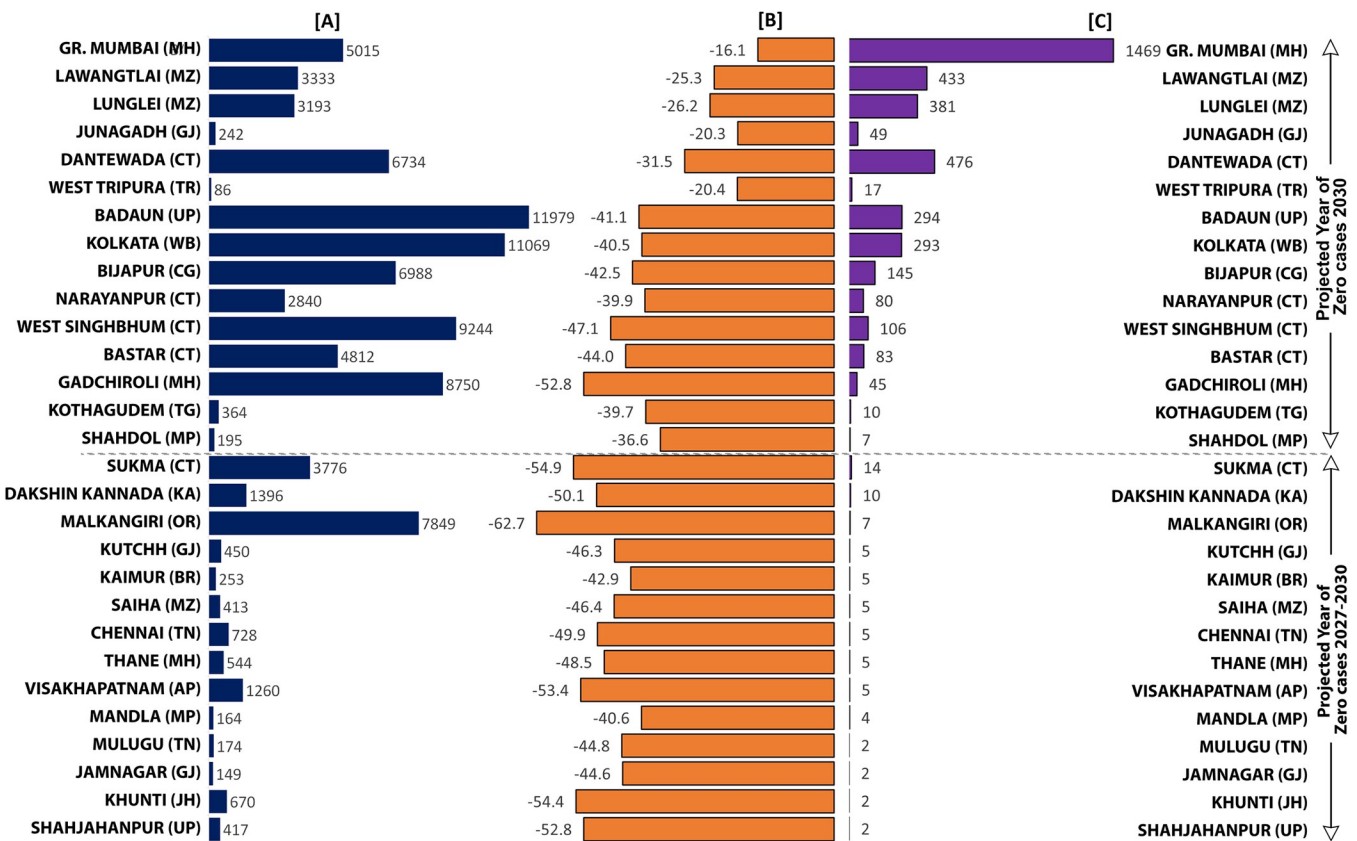

**Fig 3.** Distribution of malaria indicators ((A) Total malaria cases in 2020, (B) Maximum AROC in the last three years (2018–20) (C) projected case of malaria for 2027 using the maximum rate of reduction in the last three years (2018–20)) in the cohort of 15 districts of India that are likely to achieve zero malaria cases status post-2030 if they progress as per their current rate of reduction in malaria cases. **State Abbreviations:** AP-Andhra Pradesh, BR-Bihar, CT-Chhattisgarh, GJ-Gujarat, JH-Jharkhand, KA-Karnataka, MP-Madhya Pradesh, MH-Maharashtra, MZ-Mizoram, OR-Odisha, TN-Tamil Nadu, TG-Telangana, TR-Tripura, UP-Uttar Pradesh, WB-West Bengal.

Bengal) reported the highest number of malaria cases in 2020 (11,979 and 17,032 cases respectively), such that even a relatively high rate of reduction of more than 40% may not be enough to eliminate malaria by 2027 in these districts. They are projected to report 293 and 294 cases of malaria in 2027 respectively (Fig 4).

## Discussion

Since the discontinuation of WHO's Global Malaria Eradication Programme after it fell short of its original objective of global eradication due to the resurgence of global malaria in the 1970s and 1980s, calls for a global effort toward malaria elimination have gained a renewed vigour [11]. Since the start of the century, 21 countries have eliminated malaria, and 11 have obtained the WHO's malaria elimination certification after prevention of resurgence for three consecutive years [1]. India's NSP 2017–2022 set the target to eliminate malaria in category 1 districts by 2020 and category 2 districts by 2022 while bringing down the malaria API to less than 1 in category 3 districts. However, while the progress in reducing the malaria burden by more than 60% since 2017 is commendable [1], more focused and intense interventions are required to reduce the malaria burden to zero. Based on past experiences worldwide, as well as the elimination journey of other countries, achieving sustained elimination is significantly more complex than reducing the overall burden [9].

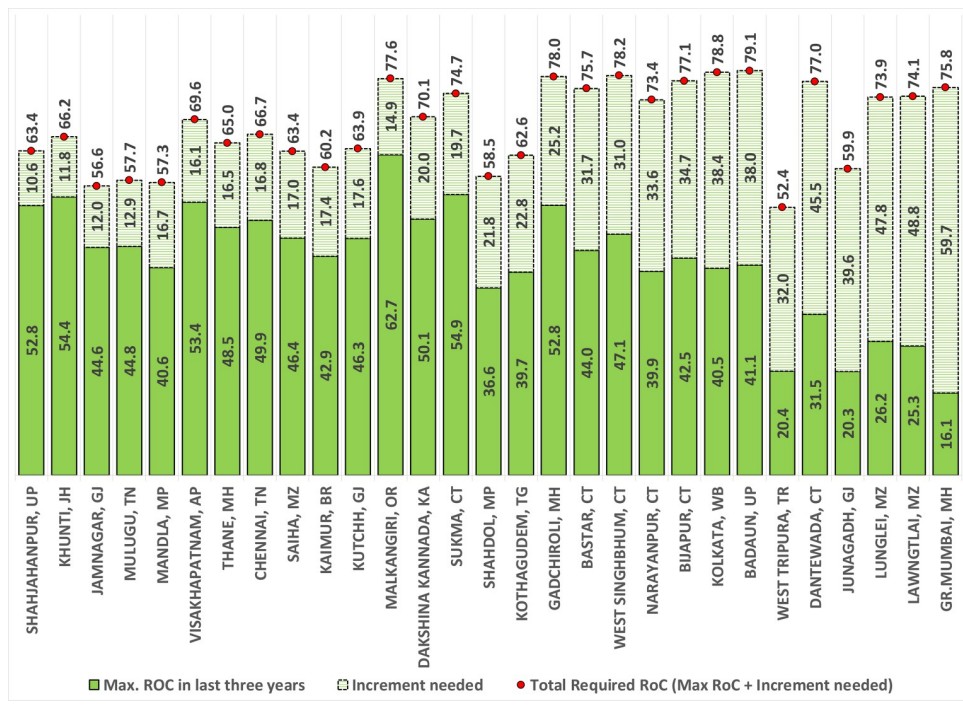

**Fig 4. Maximum AROC in the last three years, amount of increment needed in AROC and required AROC in the cohort of 15 most vulnerable districts of India that would reach zero case status after 2030 if necessary interventions are not deployed on time.**

India has significantly ramped up its intervention efforts since 2017 [7]. The results of this study reveal that almost half of the districts in India (307 districts) have either zero reported cases of malaria (117 districts) or have only negligible (<50 cases) remaining cases (205 districts) by 2020, which are likely to be reduced to zero within two-three years or so. There are ~15 districts that are crucial from a malaria elimination point of view because they may go beyond 2030 for zero case status, even considering their best reduction rate in the last three years (2018–20) (Fig 4). Excluding the urban centers of Greater Mumbai and Kolkata, all of these districts have sizeable tribal populations and belong to the states of Chhattisgarh, Jharkhand, Uttar Pradesh, Odisha, Madhya Pradesh, Tripura, and West Bengal. Even though tribals living in forested areas constitute only 6.6% of the total population in India, they contributed to almost 21% of the total malaria cases in 2019 [12].

A comparison of the recent trends in malaria incidence in these districts reveals contrasting features that contribute to sustained transmission and/or low AROC. In Greater Mumbai (Maharashtra), Junagadh (Gujarat) and West Tripura (Tripura), significant intervention efforts have brought down the overall malaria burden by almost 94%, 84% and 95% respectively in these districts between 2010 and 2019. However, the initial gains in malaria elimination have stagnated over the past 3–4 years, and the districts have witnessed an average reduction of only 5%, 9% and 10.25% since 2017 (Fig 5). In contrast, in the Lawangtlai and Lunglei districts of Mizoram, despite a more than 70% reduction in malaria cases after 2015, an increase in malaria cases has been observed in the past couple of years. While some proportion of this increase may be attributed to the ramping up of surveillance activities in the state, more focused investigations are needed to identify the underlying causes of this rise (Fig 5). Finally, in Dantewada (Chhattisgarh), after a significant drop in malaria cases since 2010 (up to .66% by 2012), a resurgence in malaria incidence was observed after 2013. While the

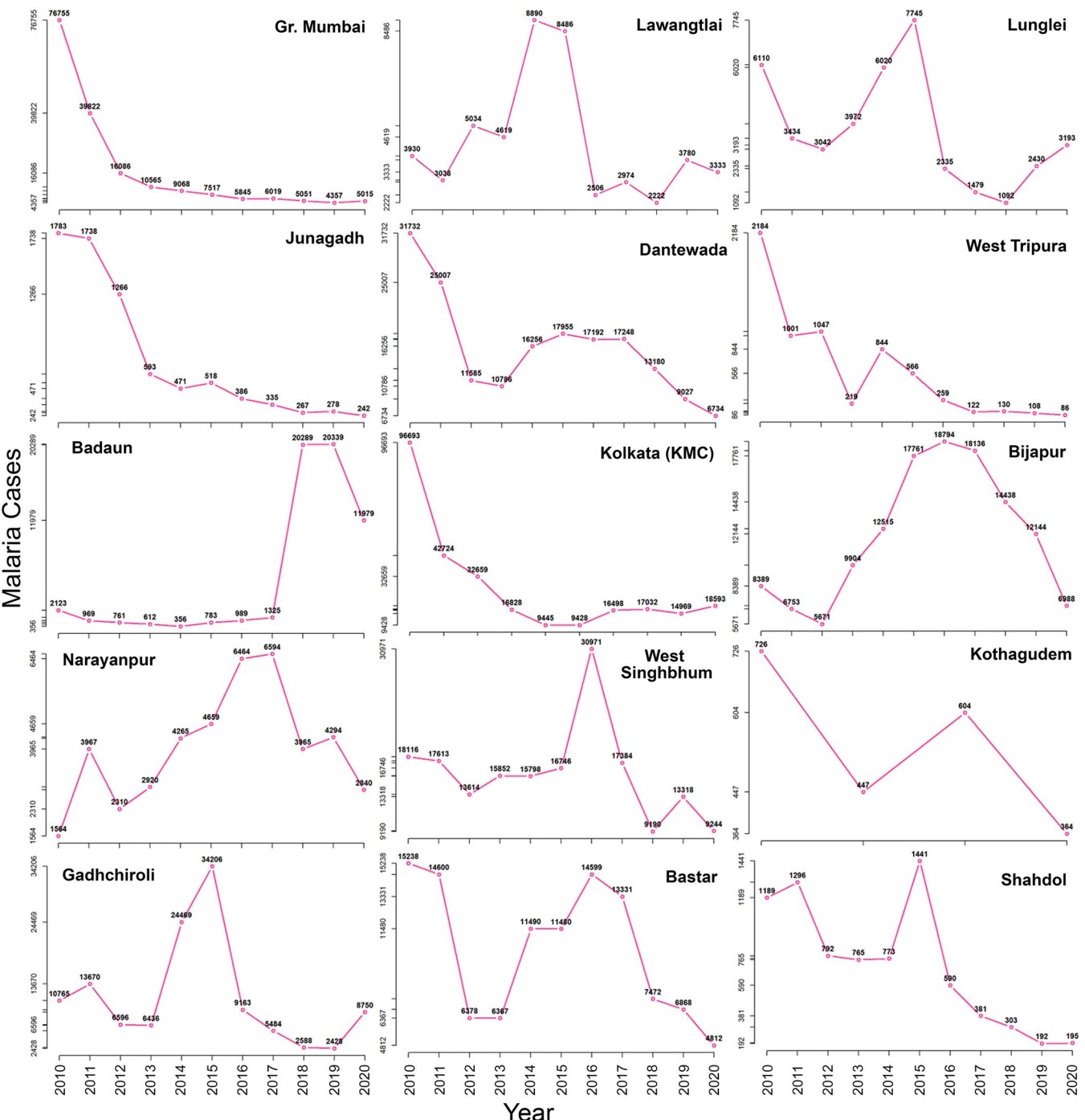

**Fig 5. Longitudinal distribution of malaria cases in the last ten years in the cohort of fifteen most vulnerable districts of India that would reach zero case status after 2030 if necessary interventions are not deployed in time.** District (State): Greater. Mumbai (MH), Lawangtlai (MZ), Lunglei (MZ), Junagadh (GJ), Dantewada (CT), West Tripura (TR), Badaun (UP), Kolkata (WB), Bijapur (CT), Narayanpur (CT), West Singhbhum (CT), Kathagudem (TG), Gadhchiroli (MH), Bastar (CT) and Shahdol (MP). **State Abbreviations:** AP-Andhra Pradesh, BR-Bihar, CT-Chhattisgarh, GJ-Gujarat, JH-Jharkhand, KA-Karnataka, MP-Madhya Pradesh, MH-Maharashtra, MZ-Mizoram, OR-Odisha, TN-Tamil Nadu, TG-Telangana, TR-Tripura, UP-Uttar Pradesh, WB-West Bengal.

number of cases has been once again brought down to less than 10,000 in the last couple of years, the current AROC is still not enough to eliminate malaria in the district by 2027 (Fig 5). Some of the districts from Uttar Pradesh and most of the districts of Bihar reported zero or very few cases in 2020. However, upon a closer inspection of the malaria data, it is observed that the Annual blood slide examination rate (ABER) in Bihar [Median (p25 to p75) = 0.065% (0.023% to 0.13%), NCVBDC-2020] is significantly lower than the recommended ~10% [10]. Disease and fever estimation surveys must be done in these regions to estimate the true burden of actual caseloads of malaria.

Besides these, India also faces several other challenges in the goal of malaria elimination, such as resistance to insecticides and drugs, complex and varied vector bionomics, logistical hurdles, inaccessible populations, natural calamities and climate change [13]. These challenges often manifest in the form of malaria outbreaks. For example, operational lapses such as poor reporting, inadequate supply of Rapid Diagnostic Kits (RDKs), and lack of an early warning signal were found to be the major contributors to a recent outbreak in the Bareilly district of Uttar Pradesh [14]. As we approach the elimination deadline, India cannot afford such lapses in its efforts toward malaria elimination. Furthermore, earlier outbreaks in Assam and Tripura have been attributed to early monsoons or changes in vector distributions [15, 16].

The current progress in malaria elimination remains commendable (85% decline in malaria cases from 2016 to 2021), and needs to be encouraged to achieve the elimination targets. Some states, namely Chhattisgarh, Madhya Pradesh, Uttar Pradesh and Jharkhand, have adopted the High Burden High Impact (HBHI) approach with support from the WHO, which aids these states in undertaking situation analysis, capacity building and finalizing district operational plans for malaria elimination [17]. Chhattisgarh has recently adopted the Malaria Mukt Bastar campaign, later expanded to the Malaria Mukt Chhattisgarh, under which mass surveillance efforts have been undertaken in several endemic regions, particularly in the tribal areas of Bastar. Uttar Pradesh has also recently launched the Dastak campaign to fight against all water and vector-borne diseases [18]. Odisha (~95% decline in malaria cases from 2016 to 2021) has been at the forefront of the malaria elimination drive in India, transitioning from the highest API state to one with the most significant reduction, driving down the overall burden of the country. The Comprehensive Case Management Programme (CCMP), a programme implemented by the state government, identified poor surveillance and asymptomatic reservoirs as the major reasons for the persistence. of the high malaria burden in the state [13] CCMP served as the basis for the implementation of the Durgama Anchalare Malaria Nirakarana (DAMaN) in 2017, which employed mass screening, treatment, and distribution of LLINs to control malaria, particularly in inaccessible tribal areas [19].

India has been on the verge of malaria elimination in the past; however, increasing insecticide resistance and urban malaria hotspots likely resulted in its resurgence [20]. As we once again approach malaria elimination, some challenges remain and we need the following endeavours (1) determining the burden of malaria through improved surveillance and involvement of the private healthcare sector [13, 21, 22], (2) increased focus on inaccessible tribal regions that are malaria endemic [12], (3) identification of asymptomatic reservoirs and submicroscopic cases of malaria [23, 24], (4) ensuring compliance to complete drug dose for *P. vivax* malaria, (5) increasing the overall coverage of LLINs [25], (6) implementation of forecasting tools and (7) use of early warning systems for malaria resource management [26]. The above tasks can be achieved by considering several solutions proposed by the National Institute of Malaria Research such as real-time data collection via a mobile app [27], data collation and visualization using digital platforms like ODK [28] and a dashboard [29, 30] which can together strengthen surveillance. Inadequate addressal of *P. vivax* malaria owing to poor compliance of primaquine can be resolved by a single dose of tafenoquine after due approvals [31].

The safety and efficacy of tafenoquine have already been established in Phase 2a/b trials in India. Tafenoquine with a companion point-of-care diagnostic device for quantitative assessment of G 6 PD deficiency can be adopted by the national programme after regulatory approvals in selected vivax-predominant areas. Novel vector control tools like Attractive Toxic Sugar Bait (ATSB) and ivermectin [32–34] can be used to circumvent the issue of emerging insecticide resistance to synthetic pyrethroids [25].

The current trends in the overall reduction of malaria in India are encouraging. Since the declaration in 2016 of the vision of malaria elimination, India has made commendable efforts in this direction. We must maintain the momentum of the downward trend of malaria cases in India. Given the diversity and size of the country, it is vital to monitor closely the malarious regions that are not performing according to the expectations and/or as per the public health investments. These recalcitrant malaria-endemic areas have the potential to remain hotspots of malaria and thus pose a challenge to the goal of malaria elimination. To support the fight against malaria, this analysis paves the way to directly address the Indian districts that are showing slower rate of decrease in malaria. This work will also encourage better use of conventional tools and the introduction of new interventional tools.

## Acknowledgments

We are very thankful to the Directorate of NCVBDC for providing data and ICMR-NIMR for all logistical support.

## Author Contributions

**Conceptualization:** Amit Sharma.

**Data curation:** Chander Prakash Yadav, Syed Shah Areeb Hussain.

**Formal analysis:** Chander Prakash Yadav, Amit Sharma.

**Investigation:** Chander Prakash Yadav, Sanjeev Gupta, Amit Sharma.

**Methodology:** Chander Prakash Yadav, Syed Shah Areeb Hussain, Amit Sharma.

**Project administration:** Chander Prakash Yadav, Sanjeev Gupta.

**Resources:** Sanjeev Gupta, Praveen K. Bharti, Manju Rahi, Amit Sharma.

**Software:** Chander Prakash Yadav, Syed Shah Areeb Hussain.

**Supervision:** Praveen K. Bharti, Manju Rahi, Amit Sharma.

**Validation:** Chander Prakash Yadav, Syed Shah Areeb Hussain, Praveen K. Bharti, Manju Rahi, Amit Sharma.

**Visualization:** Chander Prakash Yadav, Amit Sharma.

**Writing – original draft:** Chander Prakash Yadav, Syed Shah Areeb Hussain.

**Writing – review & editing:** Sanjeev Gupta, Praveen K. Bharti, Manju Rahi, Amit Sharma.

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
