## [Decision Letter · Decision Letter 0]

27 Sep 2022

PGPH-D-22-01406

Tracking malaria cases in Indian districts in the context of achieving zero indigenous case status by 2027

Dear Dr. Sharma,

Thank you for submitting your manuscript to PLOS Global Public Health. After careful consideration, we feel that it has merit but does not fully meet PLOS Global Public Health’s publication criteria as it currently stands. Therefore, we invite you to submit a revised version of the manuscript that addresses the points raised during the review process.

We look forward to receiving your revised manuscript.

Kind regards,

Emelda Aluoch Okiro

Academic Editor

Journal Requirements:

1. Please send a completed 'Competing Interests' statement, including any COIs declared by your co-authors. If you have no competing interests to declare, please state "The authors have declared that no competing interests exist". Otherwise please declare all competing interests beginning with the statement "I have read the journal's policy and the authors of this manuscript have the following competing interests:"

2. Please provide your detailed Financial Disclosure statement. This is published with the article. It must therefore be completed in full sentences and contain the exact wording you wish to be published.

a. Please clarify all sources of funding (financial or material support) for your study. List the grants (with grant number) or organizations (with url) that supported your study, including funding received from your institution. 

b. State the initials, alongside each funding source, of each author to receive each grant.

c. State what role the funders took in the study. If the funders had no role in your study, please state: “The funders had no role in study design, data collection and analysis, decision to publish, or preparation of the manuscript.”

d. If any authors received a salary from any of your funders, please state which authors and which funders.

3. Please provide separate figure files in .tif or .eps format only and remove any figures embedded in your manuscript file. Please also ensure that all files are under our size limit of 10MB.

4. In the online submission form you indicate that your data is not available for proprietary reasons and have provided a contact point for accessing this data. Please note that your current contact point is a co-author on this manuscript. According to our Data Policy, the contact point must not be an author on the manuscript and must be a third party. Please revise your data statement to a non-author institutional point of contact, such as a data access or ethics committee, and send this to us via return email. Please also include contact information for the third party organization, and please include the full citation of where the data can be found.

Additional Editor Comments (if provided):

There are some significant concerns raised by both reviewers, especially regarding ensuring sensitivity analyses are conducted around some of the methodological decisions. Two, the discussion should focus more on the Indian context and the achievable recommendations for potential solutions.

Reviewers' comments:

Reviewer's Responses to Questions

**Comments to the Author**

1. Does this manuscript meet PLOS Global Public Health’s publication criteria? Is the manuscript technically sound, and do the data support the conclusions? The manuscript must describe methodologically and ethically rigorous research with conclusions that are appropriately drawn based on the data presented.

Reviewer #1: Partly

Reviewer #2: Partly

2. Has the statistical analysis been performed appropriately and rigorously?

Reviewer #1: No

Reviewer #2: Yes

3. Have the authors made all data underlying the findings in their manuscript fully available (please refer to the Data Availability Statement at the start of the manuscript PDF file)?

Reviewer #1: No

Reviewer #2: No

4. Is the manuscript presented in an intelligible fashion and written in standard English?

Reviewer #1: Yes

Reviewer #2: Yes

5. Review Comments to the Author

Reviewer #1: Apart from the specific comments mentioned in the attached comments file, I have the following comments to the authors:

1. Do the authors think that the decline in malaria cases is a linear event? More so, when the remaining cases are very few? As per my observations, as the number of cases decease, the decline tends to be non-linear. And therefore, appropriate models should have been used for different districts based on the declining trends.

2. The toughest part on the road to zero cases is the last few cases and the authors have excluded such districts with less than 50 cases. How would you defend that?

3. Taking the last 3 years trajectory is not optimal to predict the year for achieving a zero case, specially when 2020 was badly hit by COVID. I would suggest to use a longer period of observations for making such predictions

4. How did the authors assessed the validity of their predictions?

5. Why the average of RoR was made and why not Y3-Y1 data taken into account?

6. Why wasnt a regression model used to fit the data taking the slope of the decline into account?

7. The reference numbers 24 and 28 are the same. Please use caution while reporting

8. Authors' contributions need to be mentioned in the ICMJE format

9. The figure quality is very poor and barely readable. Please provide high quality figures

10. Figure 1: it says <50 cases under exclusion and that includes 0 case as well. Please use appropriate words. The project year of elimination should be replaced with year of achieving zero case. The year range mentioned is 2027-2030 - I suppose this should be 2028-2030?. The post-2030 event should be broken down into 5-year brackets for more granularity. Why is there an arrow dropping from the Post-2030 bubble?

11. Figure 2c: The year range 2028-2030 should reflect 2027-2030? I am confused

12. Figure 3c: the legend is confusing and is incomplete. Elimination should be replaced with zero case. Please add the name of the correspond state or UT against each district mentioned.

13. Figure 4: How did you estimate the required RoR as nothing is mentioned in the methods. The required RoR cannot be depicted as a line diagram as the data is non-continuous. Please arrange the districts as per ascending/descending order of Max RoR and mention the respective states/UTs. How is it possible that a district with higher Max RoR (Vishakhapatnam) as compared to Shahjahanpur has a higher required RoR 69.6 as compared to 63.4?? Same is the case with Junagarh and West Tripura.

14. Figure 5: legend is incomplete

Reviewer #2: 1. Not all conclusions are supported by the data presented. Suggestions made in recommendation as to how to potentially make conclusions more specific to the results.

3. Not all data are available publicly. The authors state that 'most' is available - it would be useful to identify what data is available publicly and where those could be found.

6. PLOS authors have the option to publish the peer review history of their article (what does this mean?). If published, this will include your full peer review and any attached files.

**Do you want your identity to be public for this peer review?** For information about this choice, including consent withdrawal, please see our Privacy Policy.

Reviewer #1: **Yes: **Abhinav Sinha

Reviewer #2: **Yes: **Caroline Lynch

---

## [Editor Report · Decision Letter 1]

18 Nov 2022

PGPH-D-22-01406R1

Tracking malaria cases in Indian districts in the context of achieving zero indigenous case status by 2027

Dear Dr. Sharma,

Thank you for submitting your manuscript to PLOS Global Public Health. After careful consideration, we feel that it has merit but does not fully meet PLOS Global Public Health’s publication criteria as it currently stands. Therefore, we invite you to submit a revised version of the manuscript that addresses the points raised during the review process.

We look forward to receiving your revised manuscript.

Kind regards,

Emelda Aluoch Okiro

Academic Editor

Journal Requirements:

Additional Editor Comments (if provided):

The responses to the comments are satisfactory; however, the authors need to speak to these issues in the manucript as caveats or limitations, and this hasn't consistently been done.

These need to be mentioned/ discussed in the manuscript, i.e. why the authors cannot conduct any validation or sensitivity analysis and the limitation of the simplified approach- not accounting for other factors etc, the limitations of excluding cases from the private sector and the potential bias of this in their outcomes of zero malaria cases. Or the comment on suggested interventions e.g the use of TQ as a solution for patient adherence. All these need to be expanded in the discussion and not only addressed as responses to reviewer comments.
---

## [Editor Report · Decision Letter 2]

1 Dec 2022

Tracking malaria cases in Indian districts in the context of achieving zero indigenous case status by 2027

PGPH-D-22-01406R2

Dear Dr. Sharma,

We are pleased to inform you that your manuscript 'Tracking malaria cases in Indian districts in the context of achieving zero indigenous case status by 2027' has been provisionally accepted for publication in PLOS Global Public Health.

Best regards,

Emelda Aluoch Okiro

Academic Editor